# Seroprevalence and associated factors of HIV, syphilis, hepatitis B, and hepatitis C infections among sex workers in Chiangmai, Thailand during easing of COVID-19 lockdown measures

Sayamon Hongjaisee[1]*, Woottichai Khamduang[2,3,4], Nang Kham-Kjing[2,3], Nicole Ngo-Giang-Huong[3,4,5], Arunrat Tangmunkongvorakul[1]

1 Research Institute for Health Sciences, Chiang Mai University, Chiangmai, Thailand, 2 Department of Medical Technology, Faculty of Associated Medical Sciences, Chiang Mai University, Chiangmai, Thailand, 3 LUCENT international collaboration, Faculty of Associated Medical Sciences, Chiang Mai University, Chiangmai, Thailand, 4 LMI PRESTO, Faculty of Associated Medical Sciences, Chiang Mai University, Chiang Mai, Thailand, 5 Maladies Infectieuses et Vecteurs: Écologie, Génétique, Évolution et Contrôle (MIVEGEC), Agropolis University Montpellier, Centre National de la Recherche Scientifique (CNRS), Institut de Recherche Pour le Développement (IRD), Montpellier, France

* sayamon.ho@cmu.ac.th

## Abstract

During the COVID-19 pandemic, sex workers (SW) were one of the vulnerable groups affected by lockdown measures. COVID-19 had also disrupted HIV/Sexually transmitted infection (STI) testing and treatment services for sex workers due to numerous restrictions in specialist medical care. This study aims to assess the seroprevalence of HIV, syphilis, HBV, and HCV and associated factors among SW as COVID-19 restrictions were lifted. The SW aged over 18 years residing in Chiangmai, Thailand, were recruited between March and December 2022. An interview-based questionnaire was administered. Blood was collected for HIV, syphilis, HBV, and HCV serological testing. Logistic regression models were used to examine factors associated with these serological markers. Of 264 SW recruited, 52.3% were male. The median age was 31 years. Male sex workers (MSW) had higher seroprevalence of HIV (13% vs. 4.8%), syphilis (23.9% vs. 6.4%) and HCV (6.5% vs. 2.4%). Female sex workers (FSW) had higher seroprevalence of HBsAg (9.5% vs. 4.4%). A high proportion were unaware of their HIV/STI infection. MSW reporting receptive anal sex were more likely to be HIV and Treponema Ab positive. MSW reporting drug injection history were more likely to be HCV Ab positive. FSW reporting younger age at first sex were more likely to be HIV Ab positive. In conclusion, SW remains particularly affected by HIV/STIs. Despite the lockdown, HIV/STIs continued to spread, highlighting the need to provide access to HIV/STIs testing, prevention, and treatment services for this population, particularly young men.

**Data Availability Statement:** All relevant data are within the manuscript and its Supporting information files.

**Funding:** This work was financially supported by Chiang Mai University, Thailand. The funder had no role in study design, data collection and analysis, decision to publish, or preparation of the manuscript.

**Competing interests:** The authors have declared that no competing interests exist.

## Introduction

Sex workers are considered to be at high risk of acquiring and transmitting sexually transmitted infections (STIs) to their clients due to their high number of sexual partners and the high frequency of unprotected sex intercourses. The most recent estimate of the number of sex workers in Thailand is 144,000, although this is an underestimate as sex work is illegal in the country [1]. Thailand has implemented national STIs prevention policies targeting vulnerable groups, including sex workers. The Ministry of Public Health's guidelines recommend routine STIs screenings every three months and accessible treatment services at low or no cost. Collaborations with non-governmental organizations (NGOs) and community groups promote condom use, distribute preventive supplies, and raise STIs awareness within the sex worker community.

A previous study in Thailand during 2017–2019 showed that, among men who have sex with men (MSM) and transgender women (TGW) engaged in sex work, 5% were HIV Ab positive and 14% were positive for syphilis Ab [2]. An observational study conducted among MSM and TGW sex workers in Chiangmai during that period showed that 2.2% tested HIV positive [3]. Furthermore, a study conducted among migrant sex workers in Chiangmai in 2019 showed an overall prevalence of 2.3%, 3.0%, and 2.3% for HIV, syphilis, and HCV Ab respectively and 11.4% for HBsAg [4–6].

In January 2020, the coronavirus disease 2019 (COVID-19) was declared as a global public health emergency by the World Health Organization (WHO). This outbreak had a tremendous impact on the health system, social life, economy, and security of many countries. In early April 2020, the Thai government declared a state of emergency and imposed a curfew with travel and movement restrictions. A nationwide lockdown was also implemented in April-May 2020 to control the COVID-19 outbreak. Although these measures provided relief, the COVID-19 outbreak continued and lockdown measures were still in place in some provinces. COVID-19 clusters were also reported in several nightclubs in Bangkok and Chiangmai in March 2021. Emergency orders to close night entertainment venues including pubs, bars, clubs, and karaoke were still in place in June 2022.

Sex workers were one of the most vulnerable groups affected by these measures. Most had become unemployed and lost their incomes as the consequence of the lockdown and confinement measures. COVID-19 had also disrupted HIV/STI testing and treatment services for sex workers due to numerous restrictions on specialist medical care. A study of seven STI clinics in Thailand found that the number of sex workers presenting for STIs screening was drastically reduced (90%) (Department of Diseases Control, Ministry of Public Health). A previous report from Spain found that the number of reported STI cases since the start of the COVID-19 pandemic was 51% lower than expected, reaching an average of 56% during the lockdown period with a maximum reduction of 72% for chlamydia and a minimum of 22% for syphilis [7]. Another study in the US also showed evidence of significant and sustained reductions in sexual risk behaviors (number of sex partners, any substance use, and STI testing) during the pandemic [8]. The data from the Centers for Disease Control and Prevention (CDC) on Sexually Transmitted Disease (STD) surveillance 2020 showed that the reported STDs cases drastically dropped nationally during the early months of the COVID-19 pandemic in 2020. However, by the end of the year, reported cases of gonorrhoea, syphilis, and congenital syphilis surpassed their 2019 levels, indicating a continued increase in STDs [9].

During the COVID-19 pandemic response, access to HIV/ STI services was disrupted. It is unclear whether the decline in reported STI prevalence in different contexts was due to a decrease in testing or to reduced exposure or reduced sexual risk behaviors. We therefore assessed the seroprevalence of HIV, syphilis, HBV, and HCV among sex workers, in

Chiangmai, Thailand as the COVID-19 pandemic lockdown measures were gradually lifted. We also examined the factors associated with these four infections.

## Materials and methods

### Study populations

This cross-sectional study was conducted in Chiangmai, Thailand, from 1 March to 31 December 2022. In this period, lockdown and confinement measures were being lifted, but night entertainment venues were still closed until June 2022. The study included men or women who were (1) aged at least 18 years; (2) Thai or migrants; (3) sexually active with sexual partners in the past 12 months; (4) working in the entertainment venues or as freelance sex workers; (5) currently living or working in Chiangmai; (6) able to provide written informed consent. MSM, TGW, and female sex workers visiting non-governmental organization STI clinic, M Plus Foundation, were invited to participate.

### Ethical statement

The study was approved by the Human Experimentation Committee, Research Institute for Health Sciences, Chiang Mai University (Certificate of Ethical approval No. 32/2021 and No.40/2022). All participants were informed about the study and gave written informed consent. All parts of the study were kept confidential, and all participants data and laboratory results were de-identified.

### Data collection and blood collection

Participant information was collected using a questionnaire administered during a face-to-face interview by the research team and staffs from the M Plus Foundation and the Office of Disease Prevention and Control, Chiangmai, who work with sex workers and provide sexual health services in Chiangmai. Data collected included socio-demographic characteristics, health-risk behaviors, medical history, and sexual behaviors using REDCap® application. After the interview, six milliliters of peripheral blood were collected and transferred to the laboratory for processing. Plasma was separated and kept frozen at -70°C until testing for HIV, syphilis, and HCV serology and HBsAg.

### HIV, syphilis, HBV, and HCV serology testing

Plasma samples were tested with rapid immunochromatographic diagnostic tests for the detection of HIV antibodies (Determine™ HIV-1/2 test, Abbott Diagnostics, 100% sensitivity and 99.75% specificity), antibodies to *Treponema pallidum* (Determine™ Syphilis TP test, Abbott Diagnostics, 100% sensitivity and 100% specificity), HBsAg (Determine™ HBsAg 2 test, Abbott Diagnostics, 98.4% sensitivity and 99.6% specificity), and HCV antibodies (SD BIOLINE HCV, 99.3–100% sensitivity and 98.1–100% specificity). The test procedures and the interpretation of the result were carried out according to the manufacturer's instructions. All participants received an HIV test regardless of whether they already knew their HIV status.

Laboratory test results were provided to participants through the M Plus clinic. Participants testing positive for HIV and syphilis Ab were referred to the M Plus Clinic or the Office of Disease Prevention and Control, Chiangmai, for care and treatment services according to the national guidelines. Those testing positive for HBsAg, or anti-HCV received information on the prevention and management of HBV and HCV infection and were encouraged to attend the health center of their choice for clinical care.

## Data analysis

Categorical variables are presented as frequencies and percentages and continuous variables are described using the median with interquartile range (IQR). The comparison of characteristics between males and females was conducted using the Chi-square test for categorical variables and the Mann-Whitney U test for continuous variables. Univariable and multivariable logistic regression analyses were used to determine the potential association between variables (sociodemographic, health behavior, medical history, and sexual behavior) and the outcomes (HIV, syphilis, HBV, or HCV infection).

The analysis of each infection was computed independently. Variables significant in the univariable analysis with a $p$-value of <0.250 were included in the multivariable logistic regression analysis. A cutoff value of 0.250 is supported by previous study [10]. Variables that were not statistically significant with a p-value of >0.250 were removed from the model using a forward selection method, starting with no variables and adding them one by one, beginning with the one most correlated with the outcome. Odds ratios and their 95% confidence intervals (CI) were used as indicators of the strength of association. All data analyses were performed using Stata version 16.0 software (StataCorp, College Station, TX, USA). Differences were considered statistically significant when the $p$-value was <0.05.

## Results

### Characteristics of the study population

A total of 264 sex workers participated in this study. Their socio-demographic characteristics and behaviors are presented in Table 1. Approximately half (52.3%) were male. Their median age was 31 years (IQR: 25–38) overall, 27 years for men and 35.5 years for women. Most were Thai (73.5%). About 20.1% were currently living with a partner and about 70% of women had children. The median age at first sex was 17 years (IQR: 15–18). Regarding sexual behaviors, 86.2% of men reported being bisexual and 91.3% of women heterosexual. The median duration of sex work was 4 years (IQR: 2–8). Condom use was common with clients (75.6%), but not with partners (19.5%). The majority, 84.8% had a drinking habit, 48.9% had smoked and 20.8% had used recreational drug in the last 3 months. Only 16 (6.1%) had ever injected drugs. Most participants (86.4%) reported having been tested for HIV and 44.5% having been diagnosed with a genital infection (gonorrhoea, or non-gonorrhea, or syphilis, or Herpes). The majority, 85.2%, reported having body piercing or tattoos, and 40.5% reported having shared sharp objects with others.

### Seroprevalence of HIV, syphilis, HBV, and HCV

Of the 264 participants, 24 (9.1%) tested HIV Ab positive (Table 2). HIV seroprevalence was statistically higher in men than in women (13.0% vs. 4.8%, $p = 0.019$). Based on the questionnaire on previous HIV testing or knowledge of HIV status, seven men and three women were considered as newly diagnosed with HIV infection. Forty-one (15.5% of 264) sex workers tested syphilis Ab positive with a higher seroprevalence in men than in women (23.9% vs. 6.4%, $p<0.001$). Among them, 19 of 41 were newly diagnosed. Eighteen (6.8% of 264) tested positive for HBsAg (4.4% in men and 9.5% in women). Among 18 HBsAg positive cases, 11 were newly diagnosed. Twelve (4.6% of 264) were positive for HCV Ab (6.5% in men and 2.4% in women), 9 of 12 were newly diagnosed.

HIV syphilis co-infection was found in 13 (4.9%) participants, including 12 men and one woman (8.7% vs. 0.8%, $p = 0.003$). Four (1.5%) of the participants were co-infected with both HIV and HCV.

**Table 1. Characteristics of sex workers 2022 (N = 264).**

| Characteristics | | Total (%) | Male (%) | Female (%) | *p*-value* |
|---|---|---|---|---|---|
| **Sociodemographic characteristics** | | | | | |
| Sex at birth | | 264 | 138 (52.3) | 126 (47.7) | |
| Age (years) | Median | 31 (IQR: 25–38) | 27 (IQR: 22–32) | 35.5 (IQR: 31–41) | <0.001 |
| Ethnicity | Thai | 194 (73.5) | 89 (64.5) | 105 (83.3) | 0.001 |
| | Non-Thai | 70 (26.5) | 49 (35.5) | 21 (16.7) | |
| Education level | Lower than University/college | 203 (76.9) | 107 (77.5) | 96 (76.2) | 0.498 |
| | University/college | 24 (9.1) | 10 (7.3) | 14 (11.1) | |
| | None | 37 (14.0) | 21 (15.2) | 16 (12.7) | |
| Marital status | Single | 180 (68.2) | 98 (71.0) | 82 (65.1) | 0.137 |
| | Currently have a partner | 53 (20.1) | 29 (21.0) | 24 (19.0) | |
| | Separated/ Divorced/ Widowed | 31 (11.7) | 11 (8.0) | 20 (15.9) | |
| Have children | | 127 (48.1) | 39 (28.3) | 88 (69.8) | <0.001 |
| Monthly income (USD) | Median | 437 (IQR: 291–729) | 350 (IQR: 291–583) | 583 (IQR: 291–874) | <0.001 |
| Workplace | Pub/Bar/ Restaurants/ Rural road-side bar/ Cafe | 76 (28.8) | 46 (33.3) | 30 (23.8) | 0.088 |
| | Traditional massage/ Spa/Sauna | 73 (27.7) | 37 (26.8) | 36 (28.6) | 0.749 |
| | Massage parlor | 44 (16.7) | 11 (8.0) | 33 (26.2) | <0.001 |
| | Karaoke | 37 (14.0) | 15 (10.9) | 22 (17.5) | 0.123 |
| | Others | 91 (34.5) | 57 (41.3) | 34 (27.0) | 0.014 |
| **Sexual behaviors** | | | | | |
| Age at first sexual intercourse (years) | Median | 17 (IQR: 15–18) | 16 (IQR: 14–18) | 17 (IQR: 15–19) | <0.001 |
| Sexual orientation | Bisexual | 129 (48.9) | 119 (86.2) | 10 (7.9) | <0.001 |
| | Heterosexual | 119 (45.1) | 4 (2.9) | 115 (91.3) | |
| | Homosexual | 16 (6.0) | 15 (10.9) | 1 (0.8) | |
| Duration in sex work (years) | Median | 4 (IQR: 2–8) | 3 (IQR: 1–6) | 5 (IQR: 2–10) | <0.001 |
| Sexual activities (can answer >1) | Vaginal sex | 248 (93.9) | 123 (89.1) | 125 (99.2) | 0.001 |
| | Insertive anal sex | 135 (51.1) | 133 (96.4) | 2 (1.6) | <0.001 |
| | Receptive anal sex | 50 (19.0) | 31 (22.5) | 19 (15.2) | 0.188 |
| | Oral sex | 227 (86.0) | 118 (85.5) | 109 (86.5) | 0.815 |
| | Sex toys | 51 (19.3) | 29 (21.0) | 22 (17.5) | 0.465 |
| Number of sexual partners per day, in the past 3 months (persons) | Median | 1 (IQR: 1–2) | 1 (IQR: 1–1) | 1 (IQR: 1–3) | <0.001 |
| Condom use with partners, in the past month (N = 118) | All the time | 23 (19.5) | 7 (10.9) | 16 (29.6) | 0.011 |
| | Occasionally or never | 95 (80.5) | 57 (89.1) | 38 (70.4) | |
| Condom use with clients, in the past month (N = 246) | All the time | 186 (75.6) | 83 (67.0) | 103 (84.4) | 0.001 |
| | Occasionally or never | 60 (24.4) | 41 (33.0) | 19 (15.6) | |
| **Behavior characteristics** | | | | | |
| Smoking | | 129 (48.9) | 93 (67.4) | 36 (28.6) | <0.001 |
| Drinking alcohol | | 224 (84.8) | 118 (85.5) | 106 (84.1) | 0.755 |
| Recreational drug used, in the past 3 months | | 55 (20.8) | 42 (30.4) | 13 (10.3) | <0.001 |
| Ever used drug injection | | 16 (6.1) | 15 (10.9) | 1 (0.8) | 0.001 |
| **Medical history** | | | | | |
| Ever tested for HIV | | 228 (86.4) | 113 (81.9) | 115 (91.3) | 0.026 |
| HIV results (N = 228) | Positive | 14 (6.1) | 11 (9.7) | 3 (2.6) | 0.069 |
| | Negative | 209 (91.7) | 99 (87.6) | 110 (95.7) | |
| | Do not know | 5 (2.2) | 3 (2.7) | 2 (1.7) | |

*(Continued)*

**Table 1.** (Continued)

| Characteristics | | Total (%) | Male (%) | Female (%) | *p*-value* |
|---|---|---|---|---|---|
| Ever been diagnosed with | Jaundice | 3 (1.1) | 2 (1.5) | 1 (0.8) | 0.882 |
| | Liver diseases | 16 (6.1) | 7 (5.1) | 9 (7.1) | 0.233 |
| | Hepatitis B | 12 (4.6) | 4 (2.9) | 8 (6.4) | 0.063 |
| | Hepatitis C | 3 (1.1) | 2 (1.5) | 1 (0.8) | 0.306 |
| | Genital infection (Gonorrhea, or Non-gonorrhea, or syphilis, or Herpes) | 109 (44.5) | 88 (66.7) | 21 (18.6) | <0.001 |
| Ever had | Surgery or blood transfusion | 79 (30.2) | 38 (27.7) | 41 (32.8) | 0.372 |
| | needle sticked | 14 (5.3) | 11 (8.0) | 3 (2.4) | 0.129 |
| Ever received | HB vaccine | 23 (8.7) | 8 (5.8) | 15 (11.9) | 0.188 |
| **Other behaviors** | | | | | |
| Ever shared sharp objects with others | | 107 (40.5) | 65 (47.1) | 42 (33.3) | 0.023 |
| Tattoo or body piercing | | 225 (85.2) | 114 (82.6) | 111 (88.1) | 0.210 |
| Shaving in barbers (only males) | | | 68 (49.3) | | |
| Circumcision (only males) | | | 8 (5.8) | | |

*_p_-values referred to the comparison of characteristics between males and females.

## Factors associated with HIV, syphilis, HBV, and HCV infection

**Among male sex workers.** Univariable analysis of variables associated with HIV, syphilis, HBV, and HCV prevalence among male sex workers are shown in S1–S4 Tables. Variables significant in the univariable analysis ($p<0.250$) were included in the multivariable analysis.

Multivariable analysis showed that positive HIV Ab was associated with receptive anal sex (OR = 3.68, 95%CI = 1.17–11.55, $p$ = 0.026), Table 3. Positive Treponema Ab was associated with receptive anal sex (OR = 15.49, 95%CI = 4.31–55.65, $p<0.001$) and recreational drug used in the last 3 months (OR = 4.04, 95%CI = 1.33–12.27, $p$ = 0.014). HBsAg positivity was significantly associated with sexual identity (OR = 0.05, 95%CI = 0–0.53, $p$ = 0.012). HCV Ab positivity was associated with aged over 27 years (OR = 15.12, 95%CI = 1.31–174.24, $p$ = 0.029), recreational drug used in the last 3 months (OR = 11.99, 95%CI = 1.67–86.12, $p$ = 0.014), and ever having injected drugs (OR = 21.89, 95%CI = 3.45–138.79, $p$ = 0.001).

**Among female sex workers.** Univariable analysis of variables associated with HIV, syphilis, HBV, and HCV positivity among female sex workers are shown in S5–S8 Tables. Variables significant in the univariable analysis ($p<0.250$) were included in the multivariable analysis.

Multivariable analysis showed that positive HIV Ab was associated with younger age at first sex (Table 4) as compared to having first sex at older age (OR = 0.08, 95%CI = 0.01–0.59, $p$ = 0.013) and with the use of sex toys (OR = 7.37, 95%CI = 1.00–53.85, $p$ = 0.049). Positive

**Table 2. Seroprevalence of HIV, syphilis, HBV, and HCV among sex workers in Chiangmai, Thailand (N = 264).**

| Sexually Transmitted Infections | Total (N = 264) | | Male (N = 138) | | Female (N = 126) | | *P-value** |
|---|---|---|---|---|---|---|---|
| | n (%) | 95%CI | n (%) | 95%CI | n (%) | 95%CI | |
| HIV Ab | 24 (9.1) | 6.2–13.2 | 18 (13.0) | 8.3–19.8 | 6 (4.8) | 2.1–10.3 | **0.019** |
| *Treponema pallidum* Ab | 41 (15.5) | 11.6–20.4 | 33 (23.9) | 17.5–31.8 | 8 (6.4) | 3.2–12.3 | **<0.001** |
| HBsAg | 18 (6.8) | 4.3–10.6 | 6 (4.4) | 1.9–9.4 | 12 (9.5) | 5.5–16.1 | 0.096 |
| HCV Ab | 12 (4.6) | 2.6–7.9 | 9 (6.5) | 3.4–12.1 | 3 (2.4) | 0.8–7.2 | 0.107 |

*_p_-values referred to the comparison of seroprevalence between males and females.

**Table 3. Factors associated with HIV, syphilis, HBV, and HCV infections among male sex workers.**

| Characteristics | | Male | | | | |
|---|---|---|---|---|---|---|
| | | n/N (%) | Univariable | | Multivariable | |
| | | | OR (95%CI) | p-value | OR (95%CI) | p-value |
| **HIV Ab positivity** | | | | | | |
| Ever had surgery or blood transfusion | No | 16/99 (16.2) | 1.00 | | | |
| | Yes | 2/38 (5.3) | 0.29 (0.06–1.32) | **0.109** | | N.S. |
| Ever visited a community barber for shaving (males) | No | 5/70 (7.1) | 1.00 | | | |
| | Yes | 13/68 (19.1) | 3.07 (1.03–9.16) | **0.044** | | N.S. |
| Duration in sex work | < 2 years | 2/50 (4.0) | 1.00 | | 1.00 | |
| | > 2 years | 16/88 (18.2) | 5.33 (1.17–24.26) | **0.030** | 3.97 (0.83–19.02) | 0.084 |
| Receptive anal sex | No | 10/107 (9.4) | 1.00 | | 1.00 | |
| | Yes | 8/31 (25.8) | 3.37 (1.20–9.50) | **0.021** | 3.68 (1.17–11.55) | **0.026** |
| Condom use with clients, in the past month | All the time | 8/83 (9.6) | 1.00 | | 1.00 | |
| | Never or occasionally | 9/41 (22.0) | 2.64 (0.93–7.45) | **0.067** | 2.81 (0.92–8.56) | 0.069 |
| ***Treponema pallidum* Ab positivity** | | | | | | |
| Age (years) | ≤ median age (27) | 21/75 (28.0) | 1.00 | | 1.00 | |
| | > median age (27) | 12/63 (19.1) | 0.61 (0.27–1.35) | **0.222** | 0.40 (0.13–1.29) | 0.126 |
| Recreational drug used, in the past 3 months | No | 18/96 (18.8) | 1.00 | | 1.00 | |
| | Yes | 15/42 (35.7) | 2.41 (1.07–5.43) | **0.034** | 4.04 (1.33–12.27) | **0.014** |
| Ever been diagnosed with genital infections | No | 5/44 (11.4) | 1.00 | | 1.00 | |
| | Yes | 27/88 (30.7) | 3.45 (1.23–9.72) | **0.019** | 3.28 (0.86–12.54) | 0.083 |
| Age at first sexual intercourse | < 15 years old | 16/43 (37.2) | 1.00 | | 1.00 | |
| | > 15 years old | 17/95 (17.9) | 0.37 (0.16–0.83) | **0.016** | 0.37 (0.13–1.08) | 0.069 |
| Receptive anal sex | No | 18/107 (16.8) | 1.00 | | 1.00 | |
| | Yes | 15/31 (48.4) | 4.64 (1.95–11.04) | **0.001** | 15.49 (4.31–55.65) | **<0.001** |
| **HBsAg positivity** | | | | | | |
| Race | Non-Thai | 5/49 (10.2) | 1.00 | | 1.00 | |
| | Thai | 1/89 (1.1) | 0.1 (0.01–0.88) | **0.038** | 0.14 (0.01–1.35) | 0.089 |
| Sexual orientation | Heterosexual | 2/4 (50.0) | 1.00 | | 1.00 | |
| | Homosexual | 0/15 | N/A | | N/A | |
| | Bisexual | 4/119 (3.4) | 0.03 (0–0.31) | **0.003** | 0.05 (0–0.53) | **0.012** |
| **HCV Ab positivity** | | | | | | |
| Age (years) | ≤ median age (27) | 1/75 (1.4) | 1.00 | | 1.00 | |
| | > median age (27) | 8/63 (12.7) | 10.76 (1.31–88.60) | **0.027** | 15.12 (1.31–174.24) | **0.029** |
| Recreational drug used, in the past 3 months | No | 2/96 (2.1) | 1.00 | | 1.00 | |
| | Yes | 7/42 (16.7) | 9.40 (1.86–47.44) | **0.007** | 11.99 (1.67–86.12) | **0.014** |
| Ever used drug injection | No | 3/123 (2.4) | 1.00 | | 1.00 | |
| | Yes | 6/15 (40.0) | 26.67 (5.70–124.71) | **<0.001** | 21.89 (3.45–138.79) | **0.001** |

treponema Ab was associated with smoking (OR = 8.86, 95%CI = 1.59–49.26, p = 0.013), and receptive anal sex (OR = 14.51, 95% CI = 1.88–111.76, p = 0.010). HBsAg positivity was significantly associated with marital status (OR = 4.37, 95%CI = 1.20–15.60, p = 0.025). No factors were found associated with HCV Ab positivity.

## Discussion

This study found an overall high seroprevalence for HIV, syphilis, HBV, and HCV, with differences according to the sex of the participants: male sex workers had a higher seroprevalence

**Table 4. Factors associated with HIV, syphilis, HBV, and HCV infections among female sex workers.**

| Characteristics | | Female | | | | |
|---|---|---|---|---|---|---|
| | | n/N (%) | Univariable | | Multivariable | |
| | | | OR (95%CI) | p-value | OR (95%CI) | p-value |
| **HIV Ab positivity** | | | | | | |
| Drinking alcohol | No | 3/20 (15.0) | 1.00 | | 1.00 | |
| | Yes | 3/106 (2.8) | 0.17 (0.03–0.89) | **0.036** | 0.17 (0.03–1.15) | 0.070 |
| Age at first sexual intercourse | < 15 years old | 3/14 (21.4) | 1.00 | | 1.00 | |
| | > 15 years old | 3/112 (2.7) | 0.10 (0.02–0.56) | **0.009** | 0.08 (0.01–0.59) | **0.013** |
| Using sex toys | No | 3/104 (2.9) | 1.00 | | 1.00 | |
| | Yes | 3/22 (13.6) | 5.32 (1.00–28.34) | **0.050** | 7.37 (1.00–53.85) | **0.049** |
| *Treponema pallidum* Ab positivity | | | | | | |
| Smoking | No | 3/90 (3.3) | 1.00 | | 1.00 | |
| | Yes | 5/36 (13.9) | 4.68 (1.06–20.73) | **0.042** | 8.86 (1.59–49.26) | **0.013** |
| Receptive anal sex | No | 5/106 (4.7) | 1.00 | | 1.00 | |
| | Yes | 3/19 (15.8) | 3.79 (0.82–17.41) | **0.087** | 14.51 (1.88–111.76) | **0.010** |
| Oral sex | No | 3/17 (17.7) | 1.00 | | 1.00 | |
| | Yes | 5/109 (4.6) | 0.22 (0.05–1.04) | **0.057** | 0.06 (0.01–0.48) | **0.008** |
| **HBsAg positivity** | | | | | | |
| Marital status | Single | 6/82 (7.3) | 1.00 | | 1.00 | |
| | Has a partner | 1/24 (4.2) | 0.55 (0.06–4.81) | 0.590 | | |
| | Separated/divorced/widowed | 5/20 (25.0) | 4.22 (1.14–15.64) | **0.031** | 4.37 (1.20–15.60) | **0.025** |
| **HCV Ab positivity** | | | | | | |
| Drug used, in the past 3 months | No | 1/113 (0.9) | 1.00 | | | |
| | Yes | 2/13 (15.4) | 20.36 (1.71–242.94) | **0.017** | | N.S. |

for HIV (13% vs. 4.8%), syphilis (23.9% vs. 6.4%) and HCV (6.5% vs. 2.4%), while women had higher HBsAg seroprevalence (9.5% vs. 4.4%). These seroprevalences are much higher than those reported in pre-pandemic studies [2–6] in Thailand during 2017–2019 i.e. 2–5% HIV and 3–14% syphilis among MSM, TGW, and migrants engaged in sex work [2–6]. Importantly, we observed cases newly diagnosed with HIV (10 of 24 HIV Ab positive, 41.7%), syphilis (19 of 41 Treponema Ab positive, 46.3%), HBV (11 of 18 HBsAg positive, 61.1%), and HCV (9 of 12 HCV Ab positive, 75%) infections. These findings may indicate that these infections are still being detected during the relaxation of the COVID-19 quarantine measures. Although entertainment venues were closed during the COVID-19 pandemic, some sex workers continued to sell sex and became non-venue-based sex workers or freelancers. They found creative ways to continue offering personal services through many social internet platforms to find their own clients. In this study, 34.5% of sex workers were freelance or non-venue based. Unlike venue-based sex workers who were required to be tested for HIV/STI every 3 months, non-venue-based sex workers are not required to undergo regular STI or HIV testing. Previous study has also shown a significant decrease in HIV/STIs visits and testing rates during the COVID-19 pandemic [11]. In addition, access to HIV/STI testing and treatment was also difficult and limited during the COVID-19 pandemic due to numerous restrictions on specialist medical care. These factors may increase the risk of exposure to HIV and also to other STIs.

Although HCV is usually transmitted via contact with the blood of an infected person, it can also be transmitted through sexual activity. Having multiple partners, having a STI, or engaging in an anal sex appear to increase the risk of exposure to HCV. During the COVID-19 pandemic, we found that the prevalence of HCV among sex workers in Chiangmai, Thailand

was higher than that reported in 2019, at 4.6% and 2.3%, respectively. Another similar study conducted in Ethiopia in 2021 reported a higher rate of HCV (6.7%) [12]. In addition, a study in Spain showed that COVID-19 has hindered HCV and HIV screening, particularly in marginalized groups. The authors assessed the prevalence of HCV during COVID-19 vaccination in a center for addiction services (CAS) in Barcelona and a mobile testing unit (MTU) in Madrid, Spain. A high prevalence of HCV was found with 31.5% of CAS participants being positive and 14.9% of MTU participants being positive [13].

In the present study, HBsAg seroprevalence was higher in women 9.5% vs. 4.4%, although not statistically different. This difference may be due to the age of women (35.5 years (IQR: 31–41) vs. 27 years (IQR: 22–32)) who were born before the introduction of the Expanded Program on Immunization and the provision of universal neonatal HB vaccine.

We also analyzed the factors associated with these four STIs based on gender. Male sex workers who reported having receptive anal sex were at high risk of HIV and syphilis infection. This finding is consistent with other studies reporting that receptive anal sex is the riskiest sexual behavior for HIV and syphilis infection [14, 15]. Drug abuse was one of the factors associated with syphilis infection among male sex workers as reported in other epidemiological studies of syphilis in Brazil [16, 17]. Moreover, other studies showed that drug use and receptive anal sex were also associated with syphilis [15, 18]. Selling or exchanging sex was also significantly associated with alcohol and drug use and especially with sex-related drugs such as amphetamine-type stimulants (ATS), methamphetamine, poppers, or Viagra [2]. In our study, 20.8% of participants were drug users, 56% of them were ATS users, demonstrating that drug use is widely associated with involvement in sex work. Drug use, especially ATS has appeared as a problem among sex workers in several countries [19]. Previous studies have also shown that the use of methamphetamine is associated with HIV and STI incidence [20, 21] as people who use drugs are more likely to engage in unsafe sexual behaviors (condomless sex, multiple sex partners, exchange of sex for drug), which put them at higher risk of STIs.

We found that male sex workers who has aged over 27 years old were more likely to be HCV Ab positive than those aged under 27 years. This finding is consistent with previous studies which have shown that older age is significantly associated with HCV infection [22, 23]. This may be explained by the fact that older adults may engage in risky behaviors such as tattoos, blood transfusions, or unclean injections which may increase the risk of HCV infection. In addition, our results showed that the risk of HCV was higher among male sex workers who reported injecting drug use and recreational drug used. Injection drug use is an important risk factor for HCV transmission because of the sharing of used needles and injecting equipment. The prevalence of HCV among drug injectors is generally high [24]. Our findings are consistent with those of previous studies [25, 26], which have shown that sex workers especially those who inject drugs are at high risk not only for HIV but also for HCV infection and transmission. We also found that male sex workers who reported recreational drug use were at high risk of both HCV and syphilis infection.

Among female sex workers, we found that those who had their first sexual intercourse at over 15 years were less likely to be HIV Ab positive than those who had their first sexual intercourse before 15 years of age. This is consistent with previous studies showing that early sexual debut is associated with factors that may increase a young person's risk of HIV infection [27, 28]. One hypothesis to explain this may be that young sex workers have been exposed to multiple partners and are engaged in risk behaviors for HIV infection. We also found that female sex worker who smoke have a higher risk of syphilis infection. This is also in line with the report from China, which showed that current smoking and duration and frequency of smoking were associated with a higher risk of syphilis infection [29]. Smoking may be a surrogate marker of high risk behaviors, in particular increased sexual activity and less condom use [30].

In addition, the prevalence of HBV was higher in divorced women with a significant association which may related to age. This is consistent with another study showing that marital status is also associated with HBV infection [31].

This study has some limitations. First, the number of participants, the recruitment of participants was slow and challenging due to the closure of several entertainment venues in response to national restrictions during the COVID-19 pandemic. Second, collection of data was based on self-reported measures through face-to-face interviews where responses may have been influenced by social desirability bias. Third, confirmatory testing for HIV and syphilis was not conducted; however, the participants who tested positive for HIV or syphilis were referred to the M Plus Clinic or the Office of Disease Prevention and Control, Chiangmai, for confirmation and treatment according to the national guidelines.

## Conclusion

This study documents the high seroprevalence of HIV, syphilis, and HCV among sex workers in Chiangmai, Thailand when nationwide COVID-19 restrictions were gradually lifted. Sex workers, especially young men, remain disproportionally affected by HIV and STIs. Despite the lockdown measures, new cases of HIV and syphilis infection were diagnosed, indicating that HIV and syphilis continue to spread and highlighting the need to provide access to HIV/ STIs testing, prevention, and treatment services for this high-risk population. Other interventions such as increasing condom distribution, providing information on modes of transmission and methods of preventing the spread of disease should be provided. Innovative interventions such as mobile services, and outreach activities may be more useful than testing centers to reach non-venue-based sex workers or freelance workers for early HIV/STI detection and linkage to treatment and care.

## Supporting information

**S1 Table. Factors associated with HIV Ab positivity among male sex workers.**
(PDF)

**S2 Table. Factors associated with *Treponema pallidum* Ab positivity among male sex workers.**
(PDF)

**S3 Table. Factors associated with HBsAg positivity among male sex workers.**
(PDF)

**S4 Table. Factors associated with HCV Ab positivity among male sex workers.**
(PDF)

**S5 Table. Factors associated with HIV Ab positivity among female sex workers.**
(PDF)

**S6 Table. Factors associated with *Treponema pallidum* Ab positivity among female sex workers.**
(PDF)

**S7 Table. Factors associated with HBsAg positivity among female sex workers.**
(PDF)

**S8 Table. Factors associated with HCV Ab positivity among female sex workers.**
(PDF)

## Acknowledgments

The authors would like to acknowledge all participants who were involved in this study. We also thank M Plus Foundation and the Office of Disease Prevention and Control, Chiangmai staff members for their support and cooperation during the recruitment of participants and data collection.

## Author Contributions

**Conceptualization:** Sayamon Hongjaisee, Arunrat Tangmunkongvorakul.

**Data curation:** Sayamon Hongjaisee, Woottichai Khamduang, Nicole Ngo-Giang-Huong.

**Formal analysis:** Sayamon Hongjaisee, Woottichai Khamduang.

**Investigation:** Sayamon Hongjaisee, Woottichai Khamduang, Nicole Ngo-Giang-Huong.

**Methodology:** Sayamon Hongjaisee, Nang Kham-Kjing.

**Project administration:** Sayamon Hongjaisee.

**Resources:** Sayamon Hongjaisee, Arunrat Tangmunkongvorakul.

**Validation:** Sayamon Hongjaisee, Woottichai Khamduang, Nicole Ngo-Giang-Huong.

**Visualization:** Sayamon Hongjaisee, Woottichai Khamduang.

**Writing – original draft:** Sayamon Hongjaisee, Woottichai Khamduang.

**Writing – review & editing:** Sayamon Hongjaisee, Woottichai Khamduang, Nang Kham-Kjing, Nicole Ngo-Giang-Huong, Arunrat Tangmunkongvorakul.

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
