## [Decision Letter · Decision Letter 0]

22 Nov 2024

PONE-D-24-46408Seroprevalence and associated factors of HIV, Syphilis, Hepatitis B, and Hepatitis C infections among sex workers in Chiangmai, Thailand during easing of COVID-19 lockdown measuresPLOS ONE

Dear Dr. Hongjaisee,

Thank you for submitting your manuscript to PLOS ONE. After careful consideration, we feel that it has merit but does not fully meet PLOS ONE’s publication criteria as it currently stands. Therefore, we invite you to submit a revised version of the manuscript that addresses the points raised during the review process.

Your manuscript was reviewed by two experts in the field. Both found many important problems in your submission. Please review the attached comments and provide point-by-point responses.

We look forward to receiving your revised manuscript.

Kind regards,

Yury E Khudyakov, PhD

Academic Editor

PLOS ONE

2. Thank you for stating the following financial disclosure:  [Chiang Mai University].

Additional Editor Comments (if provided):

Reviewers' comments:

Reviewer's Responses to Questions

**Comments to the Author**

1. Is the manuscript technically sound, and do the data support the conclusions?

Reviewer #1: Yes

Reviewer #2: Partly

2. Has the statistical analysis been performed appropriately and rigorously? 

Reviewer #1: I Don't Know

Reviewer #2: Yes

3. Have the authors made all data underlying the findings in their manuscript fully available?

Reviewer #1: Yes

Reviewer #2: Yes

4. Is the manuscript presented in an intelligible fashion and written in standard English?

Reviewer #1: Yes

Reviewer #2: Yes

5. Review Comments to the Author

Reviewer #1: This study attempts to provide an overview regarding the prevalence of HIV, Syphilis, HBV and HCV infections among sex workers in Chiangmai, Thailand. The study well describes the current status (seroprevalence) of these infections in this specific group, including prevalent risk factors and routes of transmission (younger age at first sex for female sex workers / receptive anal sex and injection drug use for male sex workers). The authors also give their comments to the related results, refer to similar previous studies in other countries and make a good comparison between Thailand and other countries. The authors conclude that despite the current data regarding the prevalence of these infections among sex workers in Thailand, there is still a gap where additional researches are needed in order to optimize the control and prevention of sexually transmitted infections. Overall, I found that the authors have provided a high-quality study and given a substantial amount of data helping to understand better the current epidemiology status of sexually transmitted infections in Thailand. The findings of this study contribute further evidence for informing reforms to the national guidelines for the management, prevention, and control of sexually transmitted infections. The following comments need addressing:

1. The authors should address the availability of national policies or guidelines for prevention and control of sexually transmitted infections among sex workers population in Thailand.

2. Since plasma samples were tested with rapid immunochromatographic diagnostic tests, the authors should provide the sensitivity and specificity of tests.

Reviewer #2: The authors estimated the seroprevalence of several STIs among sexual workers in Thailand by recruiting participants during COVID-19. The authors also assessed the potential factors that were linked to a higher seroprevalence of STIs. Below are some suggestions to help improve the content and scientific presentation of the study.

1. Lines 91-93. The recruitment period was from 1 Mar 2022 to 31 Dec 2022. “In this period, lockdown and confinement measures were being lifted, but night entertainment venues were still closed until June 2022.” The information referred to the situation that no lockdown and confinement measures throughout the recruitment period while night entertainment venues were still closed until June 2022. That is, the closure of “night entertainment venues” was the only measure relating to COVID-19 during the recruitment period. In this case, the authors are suggested to compare and/or discuss the seroprevalence of STIs and the associated factors before and after night entertainment venues were “resumed”. This is to respond to the title for the changes “during easing of COVID lockdown measures”.

2. Please include sample size estimation. If no sample size estimation was done before the study, please justify or provide the reason.

3. Lines 139-142. Multivariable logistic regression analyses.

3.a. Cite the reference(s) and the justification for using p at 0.250 as the threshold to pick variables in the multivariable model.

3.b. “Variables that were not statistically significant with a p-value of >0.250 were removed from the model using a forward selection method.” Please elaborate on how to “remove” a variable when using a “forward selection method”.

4.a. Please change “p=0.000” to “p<0.001” throughout the text and the table.

4.b. Be consistence about the decimal places used for p-values. P-values for seroprevalence comparison on lines 166-177 were based on 3 decimal places while only 2 decimal places were presented for p-values for factor associations on lines 183-213.

5. Table 1. Include the p-value for comparing each characteristic between males and females, using the appropriate statistical tests. State the statistical tests that will be used in “Data analysis”.

6. Table 2. Please indicate whether the p-values referred to the comparison of seroprevalence between males and females. Include the total number of participants of each group (Total, Male, and Female) in the header row.

7. Please explain or justify the thresholds used to categorize continuous variables into discrete groups in the regression analyses. For example, in lines 266-267, the authors discussed that male sexual workers who were over 27 years old were more likely to be HCV Ab positive. Why did they use 27 years old at the threshold/cutoff but not 30 or 25?

8. Please also convert or include the monetary values into USD.

6. PLOS authors have the option to publish the peer review history of their article (what does this mean?). If published, this will include your full peer review and any attached files.

Reviewer #1: **Yes: **Fatemeh Farshadpour

Reviewer #2: No

---

## [Author Response · Author response to Decision Letter 0]

12 Dec 2024

Thank you very much for taking the time to review this manuscript and for giving us the opportunity to clarify and improve our manuscript. Please find the attached files which provide point-by-point responses and the corresponding revisions/corrections highlighted in the re-submitted manuscript.

---

## [Editor Report · Decision Letter 1]

16 Dec 2024

Seroprevalence and associated factors of HIV, Syphilis, Hepatitis B, and Hepatitis C infections among sex workers in Chiangmai, Thailand during easing of COVID-19 lockdown measures

PONE-D-24-46408R1

Dear Dr. Hongjaisee,

We’re pleased to inform you that your manuscript has been judged scientifically suitable for publication and will be formally accepted for publication once it meets all outstanding technical requirements.

Kind regards,

Yury E Khudyakov, PhD

Academic Editor

PLOS ONE
---

## [Editor Report · Acceptance letter]

19 Dec 2024

PONE-D-24-46408R1 

PLOS ONE

Dear Dr. Hongjaisee, 

I'm pleased to inform you that your manuscript has been deemed suitable for publication in PLOS ONE. Congratulations! Your manuscript is now being handed over to our production team.

Kind regards, 

on behalf of

Dr. Yury E Khudyakov 

Academic Editor

PLOS ONE